# Extracellular Vesicles Are Conveyors of the NS1 Toxin during Dengue Virus and Zika Virus Infection

**DOI:** 10.3390/v15020364

**Published:** 2023-01-27

**Authors:** Daed El Safadi, Grégorie Lebeau, Alisé Lagrave, Julien Mélade, Lauriane Grondin, Sarah Rosanaly, Floran Begue, Mathilde Hoareau, Bryan Veeren, Marjolaine Roche, Jean-Jacques Hoarau, Olivier Meilhac, Patrick Mavingui, Philippe Desprès, Wildriss Viranaïcken, Pascale Krejbich-Trotot

**Affiliations:** 1Unité Mixte Processus Infectieux en Milieu Insulaire Tropical (PIMIT), Université de la Réunion, INSERM U1187, CNRS UMR 9192, IRD UMR 249, Plateforme Technologique CYROI, 97490 Saint-Denis de La Réunion, France; 2Unité Mixte Diabète Athérothrombose Thérapies Réunion Océan Indien (DéTROI), Université de la Réunion, INSERM, UMR 1188, Plateforme Technologique CYROI, 97490 Saint-Denis de La Réunion, France

**Keywords:** flavivirus, Dengue, Zika, NS1, exosome, viral toxin vectorization

## Abstract

Extracellular vesicles (EVs), produced during viral infections, are of emerging interest in understanding infectious processes and host–pathogen interactions. EVs and exosomes in particular have the natural ability to transport nucleic acids, proteins, and other components of cellular or viral origin. Thus, they participate in intercellular communication, immune responses, and infectious and pathophysiological processes. Some viruses are known to hijack the cell production and content of EVs for their benefit. Here, we investigate whether two pathogenic flaviviruses i.e., Zika Virus (ZIKV) and Dengue virus (DENV2) could have an impact on the features of EVs. The analysis of EVs produced by infected cells allowed us to identify that the non-structural protein 1 (NS1), described as a viral toxin, is associated with exosomes. This observation could be confirmed under conditions of overexpression of recombinant NS1 from each flavivirus. Using different isolation methods (i.e., exosome isolation kit, size exclusion chromatography, Polyethylene Glycol enrichment, and ELISA capture), we showed that NS1 was present as a dimer at the surface of excreted exosomes, and that this association could occur in the extracellular compartment. This finding could be of major importance in a physiological context. Indeed, this capacity of NS1 to address EVs and its implication in the pathophysiology during Dengue or Zika diseases should be explored. Furthermore, exosomes that have demonstrated a natural capacity to vectorize NS1 could serve as useful tools for vaccine development.

## 1. Introduction

In the last few decades, the emergence or re-emergence of RNA viruses has led to an increased number of epidemics able to expand rapidly over large areas. Arboviruses are a particular growing threat, with several medical concerns [1]. In particular, Dengue virus (DENV) and Zika virus (ZIKV) from the flavivirus genus of the Flaviviridae family, which were discovered in the mid-20th century, have gained public awareness. Both are pathogenic flaviviruses mainly transmitted by Aedes spp. mosquitoes. Climate and environmental changes that favor the expansion of these mosquitoes beyond the tropics, and the lack of validated treatments and vaccines, increase the risk that these viruses will affect more people worldwide [2,3]. Dengue virus, which exists in four serotypes, could affect more than 50% of the world’s population by 2050 [4,5]. The number of annual human infections is currently estimated at 400 million, with only 10% of febrile disease cases reported to the WHO. About 500,000 of these infections require hospitalization and more than 20,000 lead to death [6]. The infectious pathology and its evolution are polymorphic. The risk factors for disease progression to severe forms (mainly related to dengue hemorrhagic fever and dengue shock syndrome) are still under debate. The main suggested cause is related to the magnitude of viremia, which could be associated with the virus serotype or genotype, although none of the four serotypes have been clearly identified as more pathogenic than the others. A recurring hypothesis links pathogenicity to a secondary dengue event, with intervals between infections resulting in an antibody facilitation mechanism and an exacerbated inflammatory immune response that contributes to vascular leakage. Finally, the impact of host factors as well as the roles of specific viral factors have also been extensively discussed [7,8].

Regarding ZIKV, long considered harmless with very few cases of human infection, epidemics in the Pacific Islands (2007–2014) and then in Brazil (2015) have highlighted the threat that this virus represents for the human population [9,10]. During these outbreaks, an increasing number of serious outcomes of the infection, including Guillain-Barré syndrome in adults and congenital zika syndrome (CZS), have pointed out the danger of this lesser-known infectious agent [11,12,13]. In addition, evidence of maternal–fetal and sexual transmission, as well as viral persistence in certain biological fluids, have highlighted specific characteristics of ZIKV within the flavivirus genus [14]. All of these concerns have led to a global effort in ZIKV research, notably to decipher its pathophysiology and to develop a vaccine to prevent severe forms of infection. However, for both flaviviruses, the exact mechanisms behind the different pathological consequences of the infection remain poorly understood.

Among the viral factors suspected of influencing the outcome of the disease induced by these two flaviviruses, the non-structural protein 1 (NS1) is outstanding. During an infection, this viral protein circulates in the blood and is reported to act as a toxin. Briefly, DENV and ZIKV genomes encode a polyprotein processed in seven non-structural (NS) and three structural proteins. During the maturation process, NS1 is cleaved and released into the lumen of the endoplasmic reticulum, where its N-glycosylation leads to a protein whose monomer has a molecular weight of 46–55 kDa [15]. NS1 is a multifaceted glycoprotein that forms homodimers when associated with the ER lumen and organelle membranes, and hexamers that are released from infected cells after maturation in the trans-Golgi network. The levels of circulating NS1 in the bloodstream are variable, reaching up to 50 µg·mL^−1^ in the case of dengue [16], whereas during ZIKV infection the level of soluble NS1 found in the extracellular compartment is about 30 ng·mL^−1^ [17]. In both cases, NS1 detection in the serum of patients can be used for diagnosis [18,19]. NS1 is a conserved glycoprotein among flaviviruses, with structural similarity and sequence identities in the range of 53–56% between DENV-NS1 and ZIKV-NS1 [20], however, the ZIKV-NS1 protein has not been studied as extensively as DENV-NS1. During DENV infection, NS1 is known to act as a scaffolding protein, promoting the formation of vesicle packets which host the viral replication machinery [21,22]. DENV-NS1 has also been shown to interact with components from both the innate and adaptive immune systems, thereby participating in immune evasion [23,24]. The circulating form has been associated with the development of severe forms of infection due to vascular leakage [22]. DENV-NS1, described as a viral toxin, may contribute to functional alterations of the endothelium in both cytokine-dependent [25,26] and independent manners [27]. The mechanisms involved include disruption of intercellular junctions and damage of the glycocalyx [27,28]. Recently, it has also been reported that ZIKV-NS1 can alter the junctional integrity of the endothelial cells composing the blood–brain barrier [26,29]. However, the exact underlying mechanisms of the endothelial and epithelial barrier dysfunctions and the fragility observed during ZIKV and DENV infections are not fully understood.

Exosomes are widespread small extracellular vesicles (EVs) of endosomal origin, with a heterogeneous diameter ranging from 30 to 150 nm [30]. They are produced by a variety of eukaryotic cells [31] and play a crucial role, ensuring intercellular communication and maintaining biological homeostasis [32]. This is due to their ability to transport and deliver to target cells a wide variety of molecules such as nucleic acids, metabolites, lipids, and proteins [33]. Exosomes can be distinguished from other EVs, especially those produced from the plasma membrane, by their smaller size and by various markers such as tetraspanins (CD63, CD81, CD9), TSG101, and others [34,35]. Indubitably, exosomes are nowadays a matter of special concern due to their recent involvement in pathophysiological processes such as cancer [36,37], neurodegenerative diseases [38], cardiovascular disease [39], metabolic syndrome [40,41], and even fibrosis [42,43].

Additionally, links between exosomes and infection have been explored. Their role in the progression of infection seems ambivalent, both anti-infectious by promoting immune responses and activation of immune cells, and pro-infectious by transmitting molecules of the pathogen and assisting it precisely in immune escape [44]. The involvement of exosomes in the dissemination and pathogenesis of intracellular pathogens is highly questioned because of their ability to traffic from and to plasma membranes across the endosomal pathway [35]. For example, the hepatitis C virus (HCV), a member of the Flaviviridae family, can hijack exosomal biogenesis to package its own genome into intraluminal vesicles, producing pseudo-viral yet infective particles [45,46]. HCV can also incorporate its E2 protein into exosomes, leading to increased susceptibility of recipient cells [47]. The same was recently observed with viral protein and RNA uptake by exosomes during the infection of epithelial cells with chikungunya virus [48]. In the case of dengue infection, exosomes have even been proposed as a possible “missing link” in the development of dengue hemorrhagic fever [49,50]. For Zika, EVs derived from ZIKV-infected endothelial cells and carrying ZIKV components may have the capacity to disturb the blood–brain barrier [51]. Despite these recent findings, much remains to be clarified about the specific characteristics of exosomes produced during infection with pathogenic flaviviruses, their ability to convey viral factors and their role in the development of virus-related diseases. Here, we wondered whether ZIKV and DENV2 infections could affect the exosomes produced in the extracellular compartment, and found that the NS1 protein is associated with these EVs. We then obtained cells stably expressing ZIKV-NS1 or DENV-NS1 to better characterize the NS1 protein association with exosomes.

## 2. Materials and Methods

### 2.1. Cell Culture, Viruses, and Infection

A549-Dual™ cells (InvivoGen, Toulouse, France, a549d-nfis), designated hereafter as A549 cells, were cultured at 37 °C under a 5% CO_2_ atmosphere in 4.5 g·L^−1^ glucose Dulbecco’s Modified Eagle’s Medium (DMEM) (PAN Biotech Dutscher, Brumath, France), supplemented with 1% or 10% of heat-inactivated fetal bovine serum (FBS) (Dutscher, Brumath, France), 100 U·mL^−1^ penicillin, 100 μg·mL^−1^ streptomycin, 2 mM L-glutamine, and 0.5 μg·mL^−1^ amphotericin B (PAN Biotech, Dutscher, Brumath, France), and supplemented with 10 μg·mL^−1^ blasticidin and 100 μg·mL^−1^ zeocin (InvivoGen, Toulouse, France). Human Embryonic Kidney, HEK 293T (CRL-3216) was cultured at 37 °C under a 5% CO_2_ atmosphere in DMEM, supplemented with 5% or 10% heat-inactivated FBS. For exosome purification, cells were grown in 1% FBS media or in Panserin 401 (PAN Biotech Dutscher, Brumath, France). The ZIKV clinical isolate PF-25013-18 (PF13) has been previously used and described [52]. The DENV2 clinical isolate, strain UVE/DENV2/2018/RE/47099 was provided by the H2020 European Virus Archive goes global (EVAg). The two flaviviruses were amplified on Vero E6 cells and titrated by plaque-forming unit assay before infection of the A549 cells at a multiplicity of infection (moi) of 5 for 48 h for ZIKV and 72 h for DENV2.

### 2.2. Flow Cytometry Assay of Infected Cells

To detect infected cells, we used the mouse anti-pan flavivirus envelope E protein mAb 4G2, produced by RD Biotech (1:1000 in PBS-BSA 1%). A549 cells were first harvested, fixed with 3.7% PFA in PBS for 10 min, then permeabilized for 5 min with PBS 1X 0.15% Triton X-100, and incubated with mAb 4G2 for 2 h, followed by 30 min of incubation with Alexa Fluor 488 anti-mouse secondary antibody (1:1000 in PBS-BSA 1%). The cells were washed with PBS and then subjected to flow cytometric analysis using Cytoflex (Beckman, Brea, CA, USA). Results were analyzed using cytExpert software, CytExpert version 2.4.

### 2.3. Plasmids and Transfection

For the expression of the recombinant ZIKV-NS1^FLAG-tag^, the construct was the one we previously designed, characterized, and used [53]. It is based on cloning of the NS1 sequence of the ZIKV BeH819015 strain into the pcDNA3.1(+) Neo plasmid. Mammalian codon-optimized gene coding NS1 of DENV2 (UVE/DENV2/2018/RE/47099) was established using *Cricetulus griseus* codon usage as reference. A sequence encoding an optimized secretory signal peptide [54] and a V5 tag were added respectively upstream and downstream of the synthetic NS1 sequence. The NS1 coding sequence was inserted upstream of an IRES that allows the translation of a puromycin resistance gene. The synthesis of the bicistronic gene sequence and its subcloning into Nhe-I and Xba-I restriction sites of the pcDNA3.1(+) Neo vector, to generate pcDNA3.1-NS1-DV2^V5-tag^, were performed by Genecust (Boynes, France). As control, using the same design, a sequence encoding the secreted embryonic alkaline phosphatase with a His tag (SEAP, Uniprot: P05187) was subcloned into the pcDNA3.1(+) Neo vector for pcDNA3.1-SEAP^HIS-tag^.

At 70–80% of cell confluence, HEK 293 cells were transfected by lipofection using Lipofectamine™ 3000 reagent (L3000-015, Invitrogen, Carlsbad, CA, USA), according to the manufacturer’s recommendations, with 2.5 μg of DNA.

In order to obtain stable HEK 293, cell lines expressing either ZIKV-NS1^FLAG-tag^, DENV2-NS1^V5-tag^ proteins, or SEAP^HIS--tag^, were further selected using geneticin or puromycin at 2 μg·mL^−1^ and maintained in culture in complete DMEM supplemented with the antibiotic at the same concentration.

### 2.4. Exosomes Sampling and Functional Assay of Interfering the Exosome Biogenesis

For comparison and optimization purposes, several techniques for extracting exosome fractions were used in our study. The presence of classical markers of exosomes, i.e., CD63 and CD81, was verified on the fractions isolated by these techniques.

#### 2.4.1. Exoeasy Maxi Kit^®^

Exosomes were purified from cell culture supernatants (CCS) using Exoeasy Maxi kit^®^ (Qiagen, 76064) according to manufacturer’s instructions. Briefly, cell culture supernatants (CCS) were collected and centrifuged for 10 min at 1000× *g*. For A549 infected cells, prior to exosome extraction and in order to inactivate the viruses, CCS were exposed to 1000 J/m^2^ of UV light for 15 min. 7 mL of buffer XBP was added to 7 mL of CCS and loaded on Exoeasy columns. Columns were centrifuged at 500× *g* for 1 min. Then, XWP buffer was added to the column followed by centrifugation at 5000× *g* for 5 min. Finally, exosomes were eluted from the column with 1 mL of XE buffer followed by centrifugation at 500× *g* for 5 min.

#### 2.4.2. Ultrafiltration and Size Exclusion Chromatography

CCS of transfected HEK 293 cells was harvested and filtered with a 0.22 µm filter. Then, CCS was submitted to ultra-centrifugation (20 min-17,000× *g* at 4 °C) using Optima L-80 XP Ultracentrifuge (Beckman Coulter). The resulting supernatant was ultrafiltrated using a Vivaflow 200 cell with a cut-off of 100,000 Da (Sartorius, VF20H4) and finally concentrated to 1 mL. Purification of exosomes has also been performed using size exclusion chromatography (SEC) on ÄKTA Pure™ chromatography system (GE Healthcare, Uppsala, Sweden) equipped with a 1 mL loop and a F9-C fraction collector. The 1 mL concentrated supernatant was injected on a double tandem Superose 6 Increase 10/300 GL column [55]. Unicorn software 7.0 was used for data collection and analysis. During SEC, absorbance at 260–280 nm, conductivity, and pH were monitored. Fractions of 1 mL were harvested and processed for further studies.

#### 2.4.3. PEG Exosome Purification

CCS was collected and centrifuged at 2000× *g* for 30 min at 4 °C to remove apoptotic bodies and cellular debris (giving pellet P1 and supernatant S1). A Polyethylene Glycol (PEG) solution was prepared as described previously [56] and added to S1. The 10% PEG-S1 mixture was left at 4 °C overnight and then centrifuged at 3000× *g* for 1 h. The pellet (P PEG) was resuspended in 11 mL of sterile PBS 1X. 1 mL was kept to perform the dot blot, and the remaining 10 mL was ultracentrifuged at 100,000× *g* 4 °C without brake at an acceleration 5 using a rotor Sw41. The pellet (P2) containing the extracellular vesicles resulting from the ultracentrifugation was resuspended in 1 mL of PBS.

#### 2.4.4. Functional Assay, Interfering the Exosome Biogenesis with GW4869

During the isolation of exosomes, co-elution of proteins may occur and constitute a source of contamination. This can lead to a misinterpretation of the association of these proteins with exosomal fractions [57]. To avoid this bias, it is usual to inhibit exosome biogenesis, in order to check the level of contamination during fractionation. We used GW4869 to inhibit exosome production. GW4869 is a strong inhibitor of neutral sphingomyelinases that, by preventing the formation of intraluminal vesicles, is able to block exosome production. Cells were treated overnight with 10 µM GW4869. In parallel, control cells were treated with the same volume of DMSO. CCS were collected and exosome fractions were prepared using the Exoeasy Maxi kit^®^.

### 2.5. Dynamic Light Scattering

The size and distribution of the exosomes were evaluated by dynamic light scattering (DLS) employing a Nano S Zetasizer (Malvern Instruments, Cambridge, United Kingdom) equipped with a light source from a 4 mW He-Ne laser (λ = 633 nm) and a scattering angle of 173°. Each sample was filtered on a 0.22 µm membrane before analysis. The results were processed and analyzed using Zetasizer software v3.30 (Malvern, UK).

### 2.6. Western Blot, Dot Blot, Antibodies and Recombinant Proteins

For western blot assays, samples were lysed with TEN buffer (0.1 M Tris-Cl pH 8.0, 0.01 M EDTA pH 9.0, and 1 M NaCl) and treated in Laemmli buffer, processed by SDS-PAGE and transferred onto nitrocellulose membrane, as previously reported [46,47]. For dot blot assays, 2.5 µL of CCS or cell extracts was spotted onto nitrocellulose membrane. Nitrocellulose membranes were first saturated for 30 min with TBS-5% milk–0.05% Tween. The blots were next incubated for 1 h with the appropriate dilution of primary antibody in TBS-2.5% milk–0.05% Tween. Primary antibodies used were, for NS1 detection, rabbit anti-Dengue Virus NS1 glycoprotein antibody [DN3] from Abcam (1:1000, ab41616), mouse anti-FLAG mAb (FLAG) (1:1000, Cell signaling technology, Danvers, MA, USA or 1:200, sc-166355, Santa-Cruz Biotechnology Inc., Santa Cruz, CA, USA), mouse anti-CD63 (1:200, sc-5275, Santa-Cruz Biotechnology Inc., Santa Cruz, CA, USA), mouse anti-CD81 (1:200, sc-23962, Santa-Cruz Biotechnology Inc., Santa Cruz, CA, USA), mouse anti-FLAG, mouse anti-V5, mouse anti-His, rat anti-albumin (1:1000, 04-100-812-E, Euromedex, Paris, France), rabbit anti-β-actin (1:1000, LF-PA0209, Abfrontier, Seoul, Korea). Anti-rabbit-coupled HRP and anti-mouse-coupled HRP antibodies were used as secondary antibodies (dilution 1:2000, Abcam, Cambridge, UK) in TBS-2.5% milk–0.05% Tween. The membranes were revealed with Pierce ECL Western blotting substrate (RPN 2232, Cytiva Life Sciences™, Fisher Scientific, Illkirch, France) and exposed with an Amersham imager 600 (GE Healthcare). Recombinant ZIKV-NS1^HIS-tag^ (strain Zika SPH2015) and DENV2-NS1^HIS-tag^ (Dengue type 2, strain New Guinea C) were from Sino Biological.

### 2.7. ELISA Capture of Exosomes

MaxiSorp™ plates were coated at 4 °C overnight with mouse anti-CD81 antibody (1:100, sc-23962, Santa-Cruz Biotechnology Inc., Santa Cruz, CA, USA). Then, saturation was processed with 100 µL/well of 1X assay diluent for 1 h under agitation. CCS of HEK293 stably expressing ZIKV-NS1^FLAG-tag^ was incubated for 2 h at room temperature. After a washing step, a mouse anti-FLAG antibody (1:100, sc-166355, Santa-Cruz Biotechnology Inc., Santa Cruz, CA, USA) was incubated for 1 h at room temperature under agitation. Finally, a goat anti-mouse-HRP antibody (ab6789, Abcam, Cambridge, UK) was added for 30 min at room temperature. Between each incubation step, wells were washed with PBS-Tween 0.05% three times. HRP activity was revealed using 50 μL of 3,3′, 5,5′-tetramethylbenzidine (TMB solution, 00-4201-56, eBioscence, Inc., San Diego, CA USA, Invitrogen) as HRP substrate, and stopped with 50 µL of HCL 0.2 N. Absorbance was read at 450 nm, with a reference at 570 nm, using a Fluostar Omega microplate reader (BMG LabTech).

### 2.8. QUANTI-Blue Assay

Briefly, the supernatant of HEK293 transfected with SEAP was harvested and exosomes were purified using Exoeasy Maxi kit^®^, as mentioned above. After purification, Secreted Alkaline Phosphatase activity of each fraction was assessed using QUANTI-blue solution (Invivogen, San Diego, CA, USA), following the manufacturer’s recommendations. Briefly, 20 µL of crude supernatant, exosomes fraction, and flowthrough were incubated with 180 µL of QUANTI-blue solution at 37 °C for 1 h. The absorbance was then read at 620 nm using a Fluorstar Omega microplate reader (BMG LabTech, Ortenberg, Germany).

### 2.9. Statistical Analyses

Unpaired t-test and ordinary one-way ANOVA tests were performed for statistical analysis using Graph-Pad Prism software version 9. Values of *p* < 0.05 were considered statistically significant. The degrees of significance as well as the test used are indicated in the figure captions as follows: * *p* < 0.05, ** *p* < 0.01,*** *p* < 0.001,**** *p* < 0.0001, ns = not significant.

## 3. Results

### 3.1. Infection with ZIKV or DENV2 Results in the Co-Elution of NS1 with Exosomes

We first wondered whether infection with ZIKV or DENV2, with described NS1 secretion by infected cells, would affect vesicles found in the extracellular compartment. We performed in vitro infection of the human A549 epithelial cells, which are widely used because of their permissiveness to both Zika [52] and Dengue viruses [58]. Cells were infected at moi 5 and cell culture supernatants (CCS) were collected at 48 h post-infection for ZIKV, and 72 h for DENV2. These time points correspond to steps in the infection kinetics at which cytopathic effects are not yet prominent, thus limiting the amount of debris and apoptotic bodies in the CCS. Infection was monitored by measuring cells labeled with a pan-flavi antibody (4G2) by flow cytometry (Appendix A). Exosomes were extracted from CCS using the Exoeasy Maxi kit^®^ from Qiagen. Quantitative and qualitative aspects of exosomes fractions were first analyzed by dot blot and immunodetection of tetraspanins (using anti-CD63) (Figure 1A). The CD63 signal obtained suggests an increase in the quantity of exosomes under infection conditions. This was already reported in the literature for exosomes produced by DENV-infected mosquito cells [59]. In addition, the dot blot revealed that NS1 was co-eluted with exosomes (Figure 1A). A western blot confirmed that DENV2-NS1 protein was present in the exosome fraction in its dimeric form (Figure 1B). DLS analysis of the exosome profiles reveals a reshape in the distribution of vesicles after infection (Figure 1C). EVs tend to have increased sizes, which is confirmed by the observation of the z-average (Figure 1C and Table 1). The DLS intensity measurement, which gives the proportions of the different vesicle populations, depending on their size (Figure 1C), shows that during infection, the smallest vesicles increase in number as well as in size. For the population of larger vesicles, their size increases slightly but their quantity decreases.

### 3.2. Overexpression of Recombinant ZIKV-NS1^FLAG-tag^ Protein and DENV2-NS1^V5-tag^ Lead to Co-Elution of NS1 with Exosome-Like Particles

To clarify the relationship between exosomes and NS1, stable expression of tagged-recombinant NS1 proteins from both ZIKV (ZIKV-NS1^FLAG-tag^) and DENV2 (DENV2-NS1^V5-tag^) was achieved by the transfection of HEK293. To ensure that any secreted protein does not end up in the exosome fraction, a stable HEK293 cell line was obtained by transfecting with a control construct allowing the expression and secretion of the secreted Alkaline Phosphatase (SEAP) protein. Heterogeneous populations of cells expressing ZIKV-NS1^FLAG-tag^, DENV2-NS1^V5-tag^, or SEAP^HIS-tag^ were selected and maintained in media supplemented with their respective selective antibiotics. Stable overexpression and secretion of the two NS1 recombinant proteins were assessed with western blot analysis of the CCS, as shown in Figure 2A. Heat treatment of the CCS protein extract from the ZIKV-NS1^FLAG-tag^ stable HEK293 cell line results in a single band with an apparent molecular weight of ~48 kDa. This may correspond to the monomeric form of the previously described ZIKV-NS1 protein [60]. Consistent with this result, in the non-heat-treated condition, the signal at ~76 kDa band may be related to a dimeric form of NS1. For DENV2-NS1^V5-tag^, heat treatment allows us to detect dimeric form at the expected apparent mass [61]. The hexameric forms could not be detected in our conditions. For the two cell lines overexpressing the NS1 of ZIKV and DENV2, respectively, western blot results indicate that the secreted NS1 protein is present in the CCS, mainly as a dimer, although we cannot exclude the presence of hexamers that we were not able to detect.

We then proceeded to the extraction of exosomes from CCS using Exoeasy Maxi kit^®^. We verified the exosome enrichment of our fractions by immunodetection of the tetraspanins (CD81 and CD63) (Figure 2B). We then investigated whether our secreted proteins were found in the exosome fractions. We found that although SEAP^HIS-tag^ was found in the supernatant (Figure 2C), where its activity was also detected by quantiblue measurement (Figure 2D), it was not found in the exosome fraction. We verified that SEAP was not contained within the vesicles as we could not detect it in the lysed exosomes (Figure 2C). When looking at the exosome fractions obtained from CCS of HEK cell lines that stably express NS1, we observe that both ZIKV-NS1 detected with the anti-flag and DV2-NS1 detected with the anti-V5 were co-eluted with exosomes (Figure 2C). Western blot analysis of the exosome protein extracts confirmed the presence of the NS1 proteins in the exosome fraction in their dimeric form (Figure 2E).

### 3.3. The Overexpressed ZIKV-NS1^FLAG-tag^ Protein Localizes to the Surface of Exosomes

To demonstrate the possible interaction between exosomes and NS1, we proceeded to isolate EVs through several approaches and for this, we worked with the HEK293 cell line stably expressing ZIKV-NS1^FLAG-tag^. First, we prepared EVs fractions by combining ultrafiltration and size exclusion chromatography (SEC) techniques. The first step of ultrafiltration and concentration was carried out on the CCS of HEK293-NS1, resulting in the presence of ZIKV-NS1^FLAG-tag^ protein in the >100 kDa fraction (Figure 3A). The >100 kDa fraction was then submitted to size exclusion chromatography. Based on preliminary assays, we decided to focus on the 13 mL to 15 mL fractions. These fractions were revealed in dot blot showing a potent signal for CD81 (Figure 3B). The presence of ZIKV-NS1^FLAG-tag^ in the 13 mL to 15 mL fractions was also investigated. Of note, using FLAG-tag directed antibodies, we showed the presence of NS1 in the exosome-like fraction of HEK293-NS1, while no signal was found in the exosome-like fraction of HEK293-SEAP (Figure 3B). Finally, a third method of collecting extracellular vesicles, based on polyethylene glycol precipitation, was applied. The exosome enrichment of the resulting pellet and the combined presence of ZIKV-NS1^FLAG-tag^ were confirmed (Figure 3C). Altogether, these results suggest a co-elution of ZIKV-NS1^FLAG-tag^ protein with exosome-like particles that were obtained by several different methods and support the hypothesis of an interaction between NS1 and exosomes.

At this point, we have been able to show that ZIKV-NS1^FLAG-tag^ overexpression results in detecting NS1 protein with the exosome-like EVs produced by the cells. The question that arises is how NS1 is associated with these EVs. In order to test whether NS1 is found on the surface of exosomes we performed an ELISA capture assay using anti-CD81 antibodies. Then, HEK293-ZIKV-NS1^FLAG-tag^ CCS was incubated on CD81 coated plates and ZIKV-NS1^FLAG-tag^ protein was detected with an anti-FLAG antibody. As shown in Figure 3D, a significant difference in absorbance was observed between CCS of the HEK293-NS1 and control HEK293. This indicates that NS1 protein is detectable and therefore present on the surface of exosomes. At last, DLS analysis of the fractions revealed that overexpression of ZIKV-NS1^FLAG-tag^, as well as DENV2-NS1^V5-tag^, produced an increase in the average size of the vesicles detected (Figure 3E and Table 2). Although the size ranges of EVs produced by HEK293 are different from those of A549, it can be noticed that there is an increase upon NS1 overexpression and secretion, as with ZIKV or DENV2 infected cells (Figure 1). It could therefore be assumed that the association of NS1 with EVs is likely to alter their characteristics and potentially their behaviors.

### 3.4. The Soluble NS1 Proteins from ZIKV or DENV2 Are Able to Associate with Vesicles Present in the Extracellular Compartment

Our above data demonstrates an association of NS1 with exosome-like vesicles following infection with ZIKV or DENV2, as well as upon overexpression of the two recombinant NS1 proteins. At this stage, however, it must be ensured that this association is not the result of a fractionation bias and that NS1 is not simply a contaminant. Numerous studies show that no matter which extraction method is used, none can completely exclude contaminants that can co-elute with exosomes. Certain protein contents may be important, especially when concentrating body fluids by ultrafiltration [62]. To rule out the possibility that the NS1 protein is co-eluting with exosomes without being associated with them, we inhibited exosome biogenesis by using GW4869. Since GW4869 is known to inhibit the viral cycle and, subsequently, the secretion of NS1 [63,64], we chose the stable HEK293 cells expressing ZIKV-NS1^FLAG-tag^ for the functional test of exosome biogenesis inhibition. Cells were treated or not with GW4869 at 10 µM for 16 h, and exosome extraction was proceeded using the Exoeasy Maxi kit^®^. To verify that incubation with GW4869 results in a decrease in exosome production, a dot blot was performed on CCS of HEK293 cells expressing ZIKV-NS1^FLAG-tag^. The CD63 signal was found to be significantly diminished, confirming a reduced quantity of exosomes in the presence of the inhibitor (Appendix A). If we examine the signal produced when detecting NS1 in the exosomal fraction of GW4869-treated cells, we find that it is reduced by almost half compared to the signal obtained with untreated cells (Figure 4A). The reduction in NS1 detection is indeed related to the reduction of exosomes extracted in the presence of the inhibitor. Therefore, the NS1 protein detected is not the resulting product of a contamination but a protein that is associated with exosomes and behaves accordingly.

We then aimed to determine whether the formation of vesicles-NS1 conjugates relates to intracellular exosome biogenesis or/and the ability of flavivirus NS1 to associate with lipid nanoparticles. Consequently, CCS samples from HEK293 cells grown in panserin were filtered using a centrifuge concentrator with a 100 kDa cutoff, and the concentrated EV-containing CCS were supplemented with 100 ng·mL^−1^ recombinant ZIKV-NS1^HIS-tag^ or DENV2-NS1^HIS-tag^ proteins, the concentrations consistent with known flavivirus NS1 antigenemia. After 30 min or 4 h of incubation, extraction of EVs by the PEG precipitation technique was performed as described in the previous section. Dot blot analysis of samples identified free and EV-NS1 conjugates (Figure 4B). The formation of EV-NS1 conjugates was observed for ZIKV-NS1^HIS-tag^ (immunodetected with the anti-His antibody), and DENV2-NS1^HIS-tag^ (immunodetected with the NS1 antibody). Such results suggest that soluble NS1 can interact extemporaneously with EV surface.

## 4. Discussion

The data on the involvement of extracellular vesicles, and in particular exosomes in infectious processes, remain limited, although are receiving increasing attention. The release of these vesicles from infected cells seems to be part of the defense mechanisms, particularly important in signaling to the immune system. The research community’s concern in these processes has revealed that many viruses use or hijack these pathways to transport some of their components, thus diversifying their modes of propagation and pathological effects. However, little is known about the exosomes produced in the case of DENV and ZIKV infection. Although the clinical patterns and development of disease differ between DENV and ZIKV infections, a functional alteration of the endothelial and epithelial barriers is the most prominent mechanism proposed to explain the characteristics of both infections. Effects linked to similar mechanisms of action of the two NS1 viral proteins are the most likely since this toxin is found circulating in both cases. Thus, the pro-inflammatory and barrier-damaging activity of DENV-NS1, as well as ZIKV-NS1, has been extensively documented [65,66].

In this study, we were, therefore, able to show that in vitro cell infection with ZIKV, as with DENV, results in the production of “exosome-like” extracellular vesicles. These exosomes had the hallmark presence of the classical markers CD63 and CD81, and were characterized by the associated presence of the viral protein NS1 (Figure 1). By obtaining HEK293 cell lines stably expressing and secreting a recombinant ZIKV-NS1^FLAG-tag^ or DENV2-NS1^V5-tag^, we were able to show that there was indeed an interaction of NS1, detected in its dimeric form, with exosome-like EVs (Figure 2 and Figure 4A). Then, we were able to state that dimers of NS1 were located on the surface of vesicles, as shown by ELISA capture assay (Figure 3). Finally, incubation of cell supernatant with a soluble recombinant NS1 protein from both DENV2 and ZIKV revealed that NS1 was subsequently found in the exosome fraction (Figure 4). This suggests that in addition to the hexameric form described as the form circulating in biological fluids, NS1 is actually able to take a new transport route through dimer association with EVs. Finally, when analyzing the EVs profiles in DLS, infection and interaction with NS1 appear to be able to reshape the distribution and the size of the vesicles (Figure 1 and Figure 3).

Exosomes released by cells through their secretory pathways compose the pool of small extracellular vesicles, which can be distinguished from other EVs by the presence of tetraspanins. They have established vehicles for conveying many types of chemical mediators between cells [35]. However, the exploitation of exosome communication pathways by viruses for their own benefit is a recent discovery [44]. Surprisingly, there are only a few data available on EVs produced during DENV infection of human cells, although several studies, which mainly focus on cellular factors, report altered mRNA, miRNA and protein content related to immune responses [49,67,68]. Thus, Mishra et al. report that both DENV infection and NS1 overexpression induce a high loading of exosomes with a cellular miRNA (miR-148) known to exert immunosuppressive activity [69]. These EVs, internalized by human microglial cells, would modulate neuroinflammation. The protective or contributory role of EVs in pathology has also been explored in the case of ZIKV infection, and available data has been extensively reviewed by Caobi et al. and Reyes-Ruiz et al. [70,71]. Again, several studies reported in these reviews reveal that EVs, especially those produced during the passage of the virus by the arthropod vector, could explain certain responses of human receptor cells and participate in endothelial vascular cell damage. EVs could also facilitate virus cell-to-cell transmission [72]. The recent work of Fikatas et al. provides crucial information on the role that EVs could play in modifying the integrity of the blood–brain barrier [51]. They show the ability of EVs to incorporate and transfer ZIKV RNA, NS1 and envelope protein to several recipient cells. They also show the ability of these EVs to disrupt the structure of human brain microvascular endothelial (hcMEC/D3) cell junctions.

With our findings, we now provide important new information, which is that EVs are coated with NS1. This raises questions about the impact that this viral toxin, present on the surface of exosomes, may have on communication between cells. Would it alter exosome tropism and uptake? Would it direct exosomes more specifically towards the endothelial surface, depending on the structure of the NS1 exposed [73]? Could this mode of transport stabilize the protein and allow it to circulate more sustainably in the body? These are all questions that will need to be clarified in the future.

Our study also revealed that the association of NS1 with EVs is more likely to occur in the extracellular environment rather than during the intracellular exosome biogenesis pathway, although such an early association cannot be totally excluded. We were indeed able to show that soluble or secreted dimeric proteins have the capacity to interact with the vesicles’ surface. This interaction would occur in all biological fluids during the acute phase of infection when the concentration of blood-circulating NS1 is maximal. This ability to interact with membrane surfaces and lipids was not entirely a surprise [74]. It has also been shown that NS1 of ZIKV has the ability to interact with endoplasmic reticulum membranes, as well as with liposomes that are remodeled by this interaction [75]. In addition, NS1 from dengue virus was shown to dock onto high-density lipoproteins through both protein-protein and lipid–protein interactions [76].

In light of these data, we propose the use of exosomes as a vector to improve NS1 delivery and presentation to the immune system. Exosomes are already used for viral antigenic vectorization, with several vaccines in preclinical and clinical phases [77,78]. Due to the ability of overexpressed ZIKV-NS1 and DENV2-NS1 proteins to be targeted to exosomes, our finding illustrates the possibility of using exosomes as a platform for NS1 presentation. To date, no vaccine against ZIKV has been commercialized and there is much uncertainty about the use of available dengue vaccines. Most of the vaccine strategies against flaviviruses are based on targeting the envelope (E) protein, because of its ability to induce preventive humoral and cell-mediated immune responses [79]. For instance, Dengvaxia^®^, the only licensed dengue vaccine currently commercialized, is a live-attenuated tetravalent dengue vaccine that contains DENV E and prM proteins from the four serotypes, in a yellow fever 17D backbone. One possible reason for the incomplete protection of this vaccine is a lack of T-cell immunity against non-structural proteins. Interestingly, a vaccine strategy targeting NS1 instead shows encouraging positive effects and prevents vascular permeability and hemorrhagic forms of the disease [80,81]. In addition, human antibodies targeting ZIKV-NS1 protect against disease development in a mouse model [82]. Thus, recombinant ZIKV-NS1 and DENV2-NS1 proteins would be considered as targets of choice for vaccine development [81,83]. Thanks to our work, we have available cell lines that stably overexpress and secrete these two NS1s. This makes it possible to use them for vaccine purposes. Moreover, the NS1 addressed on the surface of exosomes makes them an additional tool for a vaccine strategy aimed at an optimal presentation of the antigen. However, the safety of such a device will obviously have to be ensured, particularly if it is shown that the vectoring of NS1 by EVs potentiates its deleterious activity on endothelial barriers. All these points will need to be investigated in the future.

## 5. Conclusions

In conclusion, we demonstrated that NS1 secreted into the extracellular medium, following in vitro infection with DENV2 or ZIKV, or under conditions of overexpression and/or extemporaneous addition to the cell culture supernatant, led to an association of NS1 dimers on the surface of extracellular vesicles. This observation leads us to hypothesize that NS1 can be conveyed to privileged target cells and thereby act on infectious and pathological processes. This mode of transport could give the protein great stability in biological fluids. Finally, this addressing at the surface of exosomes could be used for antigen presentation in vaccine strategies targeting NS1 (Figure 5).

## Figures and Tables

**Figure 1 viruses-15-00364-f001:**
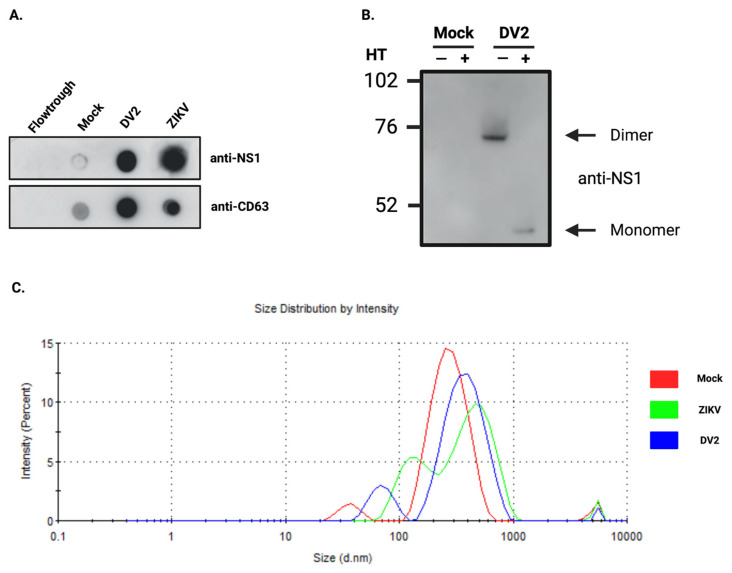
ZIKV and DENV2-infected A549 cells produce exosome-like particles associated with NS1 protein. (**A**) The NS1 protein of both ZIKV and DENV2 is found associated with the exosomal fraction obtained from the cell culture supernatants (CCS) of A549 infected cells. The tetraspanin CD63 detected on the dot blot of exosomal fractions provides insight into the quantity of exosomes extracted. (**B**) Western blot analysis of the exosome fraction, heated (HT) (+) or not (−), produced by DENV2 infected cells, show the presence of NS1 in dimeric form. A unique band with an apparent mass of ~48 kDa is observed after heating (+). (**C**) DLS analysis of exosomes produced by A549 cells during ZIKV (green) or DENV2 (blue) infection. On the y axis, the intensity gives the proportions of the different vesicle populations, depending on their size on the x axis. Both DENV2 and ZIKV infections appear to slightly reshape the distribution of vesicles.

**Figure 2 viruses-15-00364-f002:**
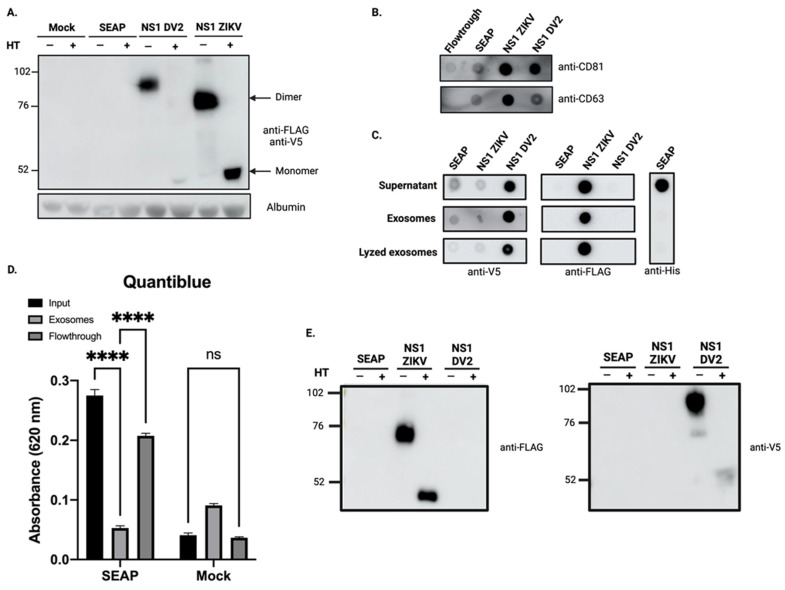
The overexpressed ZIKV-NS1^FLAG-tag^ and DENV2-NS1^V5-tag^ co-elute with exosome-like particles. (**A**) The recombinant ZIKV-NS1^FLAG-tag^ and DENV2-NS1^V5-tag^, overexpressed in stable HEK cell lines, are present in the CCS, mainly as dimers. (**B**) The tetraspanins CD81 and CD63 are immunodetected in the exosome fractions of CCS from cells expressing the different secreted recombinant proteins. (**C**) When detecting the recombinant proteins in the CCS and exosomal fractions, only the NS1 proteins were detected in the exosomal fractions, and a similar signal was obtained whether or not the vesicles had been lysed. (**D**) SEAP activity was estimated by quantiblue assay. CCS and the flowthrough remaining from the exosome extraction, but not purified exosomes, exhibited SEAP activity. Ordinary one-way ANOVA test was performed using GraphPad Prism **** *p* < 0.0001, ns (not significant) (**E**) Western blot analysis of exosome protein extracts shows that ZIKV-NS1^FLAG-tag^ and DENV2-NS1^V5-tag^ are found associated with exosomes in their dimeric form.

**Figure 3 viruses-15-00364-f003:**
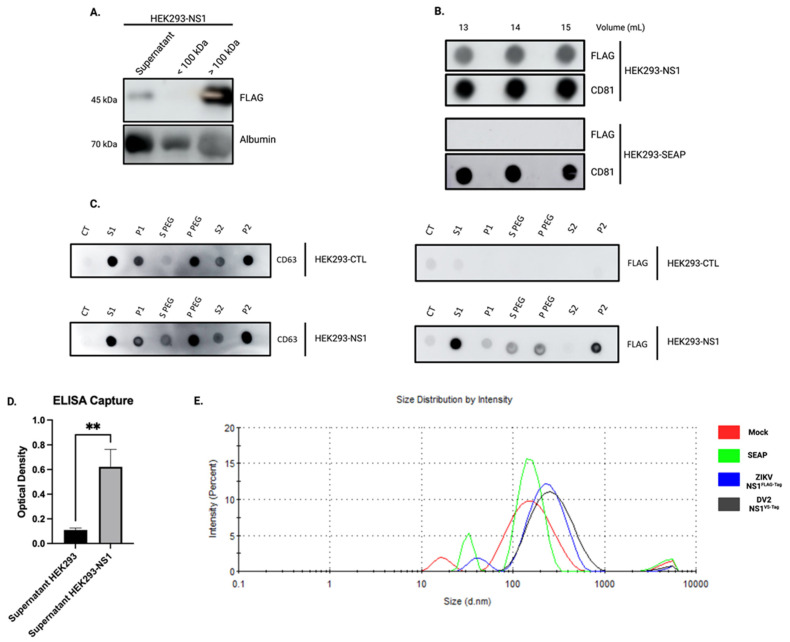
ZIKV-NS1^FLAG-tag^ is present in the exosome fraction. (**A**) Western blot analysis of HEK293-NS1 supernatant following ultrafiltration. Mouse anti-FLAG antibody was used to detect ZIKV-NS1^FLAG-tag^, while the detection of albumin served as protein loading control. (**B**) Dot blot analysis of putative exosome fractions. After the ultrafiltration step, size exclusion chromatography was carried out on the >100 kDa fraction. Detection of ZIKV-NS1^FLAG-tag^ was achieved using a mouse anti-FLAG antibody, whereas CD81 was chosen as a marker for exosomes. (**C**) CCS containing CD63-positive EVs and secreted ZIKV-NS1^FLAG-tag^ (S1) were treated with PEG and centrifuged to obtain a pellet enriched in EVs. CD63 was found in the first pellet (P PEG) and after the second ultracentrifugation step (P2), validating the enrichment of exosomes which were also found associated with ZIKV-NS1^FLAG-tag^. (**D**) ELISA capture of the exosomes in the supernatant of HEK293-ZIKV-NS1^FLAG-tag^ versus control. Exosomes were captured by an anti-CD81 antibody coated on Maxisorp^®^ plate. NS1^FLAG-tag^ protein was detected using a mouse anti-FLAG antibody. Ordinary one-way ANOVA test was performed using GraphPad Prism, ** *p* < 0.001. (**E**) DLS analysis of the exosomal fractions with the proportion of vesicles by size graph and the corresponding z-average-table (Table 2).

**Figure 4 viruses-15-00364-f004:**
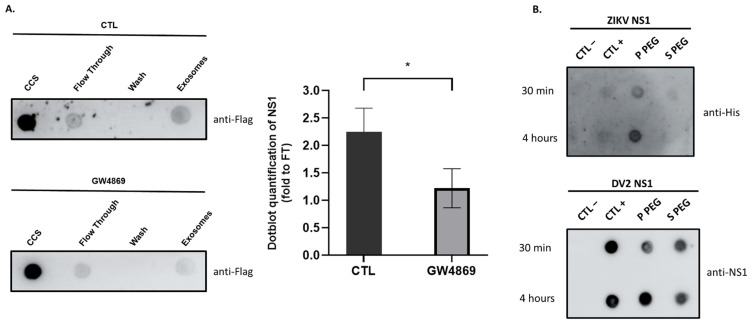
ZIKV and DV2 NS1 recombinant proteins bind to extracellular vesicles. (**A**) HEK 293 cells expressing ZIKV-NS1^Flag--tag^ protein were treated or not with GW4869 at 10 µM for 16 h in order to inhibit exosome’s biogenesis. The cell culture supernatant (CCS) was collected for exosomes extraction using the Exoeasy kit. A dot blot was performed using anti-Flag and then quantified using imageJ. The flow through (FT) is the remaining part of the first exosome extraction step, and the wash is the second flow through, considered as a negative control. The quantity of NS1 associated with exosomes was reduced when cells were treated with GW4869, thus eliminating the co-elution bias of NS1 protein with exosomes. Unpaired t-test was performed using GraphPad Prism, * *p* < 0.05. (**B**) CCS of HEK was collected 48 h post-platting and incubated with ZIKV-NS1^HIS-tag^ or DENV2-NS1^HIS-tag^ for 30 min and 4 h. CCS was then proceeded to PEG exosome precipitation. The negative control (CTL−) corresponds to panserin. The positive control (CTL+) corresponds to 0.2 ng of NS1 solubilized in panserin. Dot blots were performed using anti-His and anti-NS1 antibodies to detect ZIKV and DV2 NS1 proteins, respectively. NS1 of ZIKV and DV2 were found associated with EVs as early as 30 min post-incubation.

**Figure 5 viruses-15-00364-f005:**
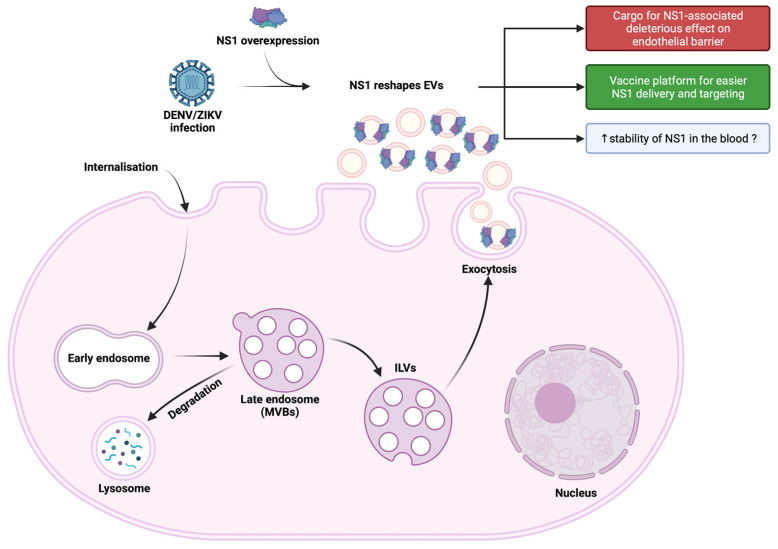
Graphical overview of the association of ZIKV and DENV NS1 proteins with exosomes.

**Table 1 viruses-15-00364-t001:** Increased size of exosome during both ZIKV and DENV2 infection.

Sample	Z-Average (d.nm)	Standard Deviation	*p*-Value (vs.mock)
Mock	230.6	3.519	−
ZIKV	259.4	16.32	0.0057
DENV2	277.5	7.301	0.0001

Z-average: The Z-average is the intensity-weighted mean hydrodynamic size of the ensemble collection of particles measured by dynamic light scattering (DLS).

**Table 2 viruses-15-00364-t002:** Overexpression of ZIKV and DENV2 NS1 lead to an increase in exosomes size.

Sample	Z-Average (d.nm)	Standard Deviation	*p*-Value (vs.mock)
Mock	123.4	8.916	−
SEAP	119.5	3.627	ns
ZIKV NS1^FLAG-Tag^	183.7	2.970	0.0001
DENV2 NS1^FLAG-Tag^	209.3	4.382	0.0001

## Data Availability

The data presented in this study are available in this paper and Appendix A.

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
