# Peer review of "Extracellular Vesicles Are Conveyors of the NS1 Toxin during Dengue Virus and Zika Virus Infection"

_viruses, 2023, doi:10.3390/v15020364_

Round 1
Reviewer 1 Report
The manuscript by Safadi et al. is a perfect match for the journal Viruses. The report on NS1 secreted into the extracellular medium, following in vitro infection with DENV2 or ZIKV, or under conditions of overexpression and/or extemporaneous led to an association of NS1 dimers on the surface of extracellular vesicles. This study shows that exosomes could be used for antigen presentation in vaccine strategies targeting NS1. While the manuscript is clear in most parts, some minor revisions, if necessary can be incorporated into the manuscript.
1. The introduction is too long, it is highly recommended to shorten it.
2. Line 125. This sentence seems to be awkward; it is better to remove it.
3. Line 232. What is 0,22? Please correct this mistake.
4. In figure 1B, there is no bot analysis for ZIKV. Please explain the logic.
Author Response
The manuscript by Safadi et al. is a perfect match for the journal Viruses. The report on NS1 secreted into the extracellular medium, following in vitro infection with DENV2 or ZIKV, or under conditions of overexpression and/or extemporaneous led to an association of NS1 dimers on the surface of extracellular vesicles. This study shows that exosomes could be used for antigen presentation in vaccine strategies targeting NS1. While the manuscript is clear in most parts, some minor revisions, if necessary can be incorporated into the manuscript.
First of all, we would like to thank the reviewer for his thorough reading of our manuscript and for his constructive comments. Here are our answers and some explanations of the reviewer’s comments.
- The introduction is too long, it is highly recommended to shorten it.
We wanted to cover as much information as possible on the infection and specificities of the Dengue and Zika viruses, especially with the singularity of NS1. In addition, we had to introduce the recent data on extracellular vesicles. Indeed, we agree that this set of background data makes the introduction extremely dense. We followed your recommendation and have therefore shortened the section describing the well-known basic characteristics of flaviviruses (lines 71 to 81 were suppressed).
- Line 125. This sentence seems to be awkward; it is better to remove it.
We rewrote the sentence (now line 119) : In the case of dengue infection, exosomes have even been proposed as a possible "missing link" in the development of dengue hemorrhagic fever.
- Line 232. What is 0,22? Please correct this mistake.
Thank you for rising this mistake. We have replaced it with 0.22 mm.
- In figure 1B, there is no bot analysis for ZIKV. Please explain the logic.
That's a very good point. Unfortunately, the only antibody we currently have in our laboratory is a monoclonal antibody ‘anti-Dengue Virus NS1 glycoprotein [DN3]’ from Abcam (ab41616). With this antibody we were able to detect NS1 on the dot blot of the exosomal fractions from both ZIKV and DENV infections and we detected DENV2-NS1 on the western blot. However, despite repeated assays with different antibody concentrations, we were unable to obtain a signal for ZIKV-NS1 on the western blot. Given the later results obtained with the recombinant proteins of each of the two viruses, it can be assumed that a dimeric form of ZIKV-NS1 would have been found.
Reviewer 2 Report
El Safadi and colleagues assessed if ZIKV and DENV2 infections affect exosome production in A549 cells and then characterized the association of NS1 protein with exosomes in HEK293 cells stably expressing NS1 from ZIKV or DENV2.
The authors report that (1) infection of A549 cells with ZIKV and DENV2, leads to an increase in the size and number of exosomes produced by the cells, moreover the DENV2-NS1 is co-eluted with the exosomes. (2) To study the relationship between exosomes and the NS1 protein, cells stably expressing tagged-recombinant NS1 protein from both viruses are obtained, having as a control cells that express tagged-SEAP (Secreted Alkaline Phosphatase protein) ensuring that any secreted protein does not end up in the exosome fraction. Then exosomes are extracted from the supernatants of the respective cells and although SEAP protein is in the supernatant, only DENV2-NS1 and ZIKV-NS1 are in the exosome fraction isolated from the supernatant present mainly as a dimer. (3) ZIKV-NS1 associates with exosomes isolated by different approaches and localizes in the surface of the exosomes. Moreover, the overexpression of NS1 of DENV2 and ZIKV leads to an increase in the average size of vesicles in the supernatant. (4) When the exosome biogenesis is inhibited the detection of NS1 diminishes. And soluble NS1 from ZIKV and DENV2 can interact with the extracellular vesicles in the extracellular compartment.
The study demonstrates that DENV2-NS1 and ZIKV-NS1 of infected cells, cells that overexpress the protein and the soluble protein can associate with extracellular vesicles. The importance of this association in pathogenesis needs to be further studied.
Comments:
· Statistical analyses, section 2.9. Please describe what kind of test were performed using Graph-Pad prism software to analyze the data.
· Results, section 3-1 starting in line 303 the authors indicate that the quantity of vesicles changes, please explain how this was assessed and how the reader can see the change in quantity in the figure.
· Section 3.3 why only ZIKV-NS1 localization in the exosomes was analyzed? Is there a reason, please discuss it.
· Figure 3B, in the control of HEK293-SEAP why anti-FLAG was used and no anti-His to demonstrate that the secreted protein is not in the EVs?
· Figure 3, please be concise with the tags of the HEK293 used as control, in 3C you use HEK293 CTL and in 3D only HEK293.
· Section 3.4 line 445 you say that the reduction of NS1 is related to reduction of exosomes extracted, however you only used anti-FLAG antibody, an antibody against a marker of exosomes should be used to observe the quantity of exosomes.
· Line 458 refers to figure 4, it should say Figure 4B.
Author Response
Reviewer #2
El Safadi and colleagues assessed if ZIKV and DENV2 infections affect exosome production in A549 cells and then characterized the association of NS1 protein with exosomes in HEK293 cells stably expressing NS1 from ZIKV or DENV2.
The authors report that (1) infection of A549 cells with ZIKV and DENV2, leads to an increase in the size and number of exosomes produced by the cells, moreover the DENV2-NS1 is co-eluted with the exosomes. (2) To study the relationship between exosomes and the NS1 protein, cells stably expressing tagged-recombinant NS1 protein from both viruses are obtained, having as a control cells that express tagged-SEAP (Secreted Alkaline Phosphatase protein) ensuring that any secreted protein does not end up in the exosome fraction. Then exosomes are extracted from the supernatants of the respective cells and although SEAP protein is in the supernatant, only DENV2-NS1 and ZIKV-NS1 are in the exosome fraction isolated from the supernatant present mainly as a dimer. (3) ZIKV-NS1 associates with exosomes isolated by different approaches and localizes in the surface of the exosomes. Moreover, the overexpression of NS1 of DENV2 and ZIKV leads to an increase in the average size of vesicles in the supernatant. (4) When the exosome biogenesis is inhibited the detection of NS1 diminishes. And soluble NS1 from ZIKV and DENV2 can interact with the extracellular vesicles in the extracellular compartment.
The study demonstrates that DENV2-NS1 and ZIKV-NS1 of infected cells, cells that overexpress the protein and the soluble protein can associate with extracellular vesicles. The importance of this association in pathogenesis needs to be further studied
First of all, we would like to thank the reviewer for the extensive reading of our manuscript. Here are our answers and some explanations of the reviewer’s comments.
- Statistical analyses, section 2.9. Please describe what kind of test were performed using Graph-Pad prism software to analyze the data.
Thank you for pointing this out. The statistical tests that have been used have been added. Section 2.9, line 272: “Unpaired t-test and Ordinary one-Way ANOVA tests were performed for statistical analysis using Graph-Pad Prism software version 9 ».
- Results, section 3-1 starting in line 303 the authors indicate that the quantity of vesicles changes, please explain how this was assessed and how the reader can see the change in quantity in the figure.
We agree and have revised it accordingly in the text and in the legend of the figure 1C (line 300 : “The DLS intensity measurement which gives the proportions of the different vesicle populations depending on their size (Figure 1C), shows that during infection, the number of the smallest vesicles increases as well as their size.”)( line 313 : “On the y axis, the intensity gives the proportions of the different vesicle populations, depending on their size on the x axis.”)
- Section 3.3 why only ZIKV-NS1 localization in the exosomes was analyzed? Is there a reason, please discuss it.
The objective of this section was to verify that several alternative methods of EVs purification would confirm that NS1 was indeed found associated with exosomes fractions. To avoid duplications and given the similarities observed previously, we chose to focus on a single representative NS1. We thus now indicate it in the text (line 366). The following paragraph (3.4), where we show that the two commercial recombinant NS1 proteins, when added to the EVs fractions, will become associated with them, confirms a shared ability of the two NS1s from ZIKV as DENV2 to interact with the vesicle membranes.
- Figure 3B, in the control of HEK293-SEAP why anti-FLAG was used and no anti-His to demonstrate that the secreted protein is not in the EVs?
This is a very good point. The absence of association of SEAP with EVs has already been demonstrated in the previous section and Figure 2C and 2D. Here, the anti-FLAG was used on the exosomal SEAP fractions to verify that the signal observed for the HEK 293-NS1 fractions was specific.
- Figure 3, please be concise with the tags of the HEK293 used as control, in 3C you use HEK293 CTL and in 3D only HEK293.
We agree and modified accordingly : HEK293 was replaced by HEK293 CTL.
- Section 3.4 line 445 you say that the reduction of NS1 is related to reduction of exosomes extracted, however you only used anti-FLAG antibody, an antibody against a marker of exosomes should be used to observe the quantity of exosomes.
We agree that it was important to verify that the inhibitor of the exosome biogenesis (GW4869) had indeed reduced the quantity of exosomes in the cell culture supernatants. We thus repeated this experiment, verified the quantitative decrease of exosomes and added this data in an additional figure (Figure S2) that you can see below, with comments line 437.
Figure S2– HEK293 cells expressing ZIKV-NS1Flag--tag protein were treated or not with GW4869 at 10µM overnight in order to inhibit exosome’s biogenesis. The cell culture supernatant (CCS) was collected. A dot blot was performed using anti-CD63 and then quantified using imageJ. The quantity of exosomes was significantly reduced when cells were treated with GW4869, thus showing that the exosome’s biogenesis was inhibited. Unpaired t-test was performed using GraphPad Prism, *p < 0.05.
- Line 458 refers to figure 4, it should say Figure 4B.
Yes, we corrected it in the new version of our manuscript.

Reviewer 3 Report
This manuscripts, the authors overexpressed the NS1, and analysis the interaction with extracellular vesicles through several assays, maybe have reference significance in flavivirus, but some improvements should be executed.
1. In this paper, the authors just executed research on DENV and ZIKV, but the title is “flavivirus infection”, I think this in is not correct.
2. In Line 167-168, “A sequence encoding a Secrecon 167 signal peptide and a V5 tag was added respectively upstream and downstream of the 168 synthetic NS1 sequence.” “was” should be changed to “were”.
3. In Line 198, the unit of 0.22 μΜ should be changed to “μm” and keep the same writing with Line 223.
4. In fig 2, the number of “2C” should be changed, and why is the fig 2E without internal reference, think this should be added. And in order to compare the expression of the protein, I think in fig 2E, the picture should put top and down together.
5. In part 3.3, why authors didn’t execute indirect or direct immunofluorescence assay to analysis the localization.

Author Response
This manuscripts, the authors overexpressed the NS1, and analysis the interaction with extracellular vesicles through several assays, maybe have reference significance in flavivirus, but some improvements should be executed.
First of all, we would like to thank the reviewer for the extensive reading of our manuscript. Here are our answers, explanations and improvements made thanks to the reviewer's comments and suggestions.
- In this paper, the authors just executed research on DENV and ZIKV, but the title is
“flavivirus infection”, I think this in is not correct.
We have taken your comment into account and changed the title accordingly by replacing flavivirus with Dengue and Zika.
- In Line 167-168, “A sequence encoding a Secrecon 167 signal peptide and a V5 tag was added respectively upstream and downstream of the 168 synthetic NS1sequence.” “was” should be changed to “were”.
We have fixed this.
- In Line 198, the unit of 0.22 μΜ should be changed to “μm” and keep the same writing with Line 223.
Thank you and we have modified it. (now lines 192 and 226)
- In fig 2, the number of “2C” should be changed, and why is the fig 2E without internal reference, think this should be added. And in order to compare the expression of the protein, I think in fig 2E, the picture should put top and down together.
We have made the requested changes. Concerning an internal reference, as the proteins are extracted from the exosomal fractions, a tetraspanin detection is indicated. We have indeed used both anti CD63 and anti CD81 which detections are presented in figure 2B and Western blot was just there to confirm the presence of the NS1 proteins in their dimeric form.
- In part 3.3, why authors didn’t execute indirect or direct immunofluorescence assay to analysis the localization.
Thank you for your suggestion. Unfortunately the size of the exosomes (30-150nm) is not compatible with fluorescence microscopy imaging to reveal a localization of NS1 on the surface of the vesicles. To overcome this difficulty, we chose to show that NS1 was present on the surface of vesicles using the Elisa capture assay and the native dot blot assay.